# Study on the Properties of Multi-Walled Carbon Nanotubes (MWCNTs)/Polypropylene Fiber (PP Fiber) Cement-Based Materials

**DOI:** 10.3390/polym16010041

**Published:** 2023-12-21

**Authors:** Xiangjie Niu, Yuanzhao Chen, Zhenxia Li, Tengteng Guo, Meng Ren, Yanyan Chen

**Affiliations:** 1School of Civil Engineering and Communication, North China University of Water Resources and Electric Power, Zhengzhou 450045, China; xiangjieniu@163.com (X.N.);; 2Technology Innovation Center of Henan Transport Industry of Utilization of Solid Waste Resources in Traffic Engineering, North China University of Water Resources and Electric Power, Zhengzhou 450045, China; 3Henan Province Engineering Technology Research Center of Environment Friendly and High-Performance Pavement Materials, Zhengzhou 450045, China

**Keywords:** multi-walled carbon nanotubes, polypropylene fiber, cement-based materials, mechanical properties, durability performance, microstructure, pore structure

## Abstract

In order to improve the mechanical properties and durability of cement-based materials, a certain amount of multi-walled carbon nanotubes (MWCNTs) and polypropylene fiber (PP fiber) were incorporated into cement-based materials. The mechanical properties of the multi-walled carbon nanotubes/polypropylene fiber cement-based materials were evaluated using flexural strength tests, compressive strength tests, and splitting tensile tests. The effects of multi-walled carbon nanotubes and polypropylene fiber on the durability of cement-based materials were studied using drying shrinkage tests and freeze–thaw cycle tests. The effects of the multi-walled carbon nanotubes and polypropylene fibers on the microstructure and pore structure of the cement-based materials were compared and analyzed using scanning electron microscopy and mercury intrusion tests. The results showed that the mechanical properties and durability of cement-based materials can be significantly improved when the content of multi-walled carbon nanotubes is 0.1–0.15%. The compressive strength can be increased by 9.5% and the mass loss rate is reduced by 27.9%. Polypropylene fiber has little effect on the compressive strength of the cement-based materials, but it significantly enhances the toughness of the cement-based materials. When its content is 0.2–0.3%, it has the best effect on improving the mechanical properties and durability of the cement-based materials. The flexural strength is increased by 19.1%, and the dry shrinkage rate and water loss rate are reduced by 14.3% and 16.1%, respectively. The three-dimensional network structure formed by the polypropylene fiber in the composite material plays a role in toughening and cracking resistance, but it has a certain negative impact on the pore structure of the composite material. The incorporation of multi-walled carbon nanotubes can improve the bonding performance of the polypropylene fiber and cement matrix, make up for the internal defects caused by the polypropylene fiber, and reduce the number of harmful holes and multiple harmful holes so that the cement-based composite material not only has a significant increase in toughness but also has a denser internal structure.

## 1. Introduction

As a traditional building material, cement-based materials play a vital role in various fields. However, their own brittleness, poor crack resistance, and other shortcomings cannot be ignored [1,2]. These shortcomings limit the application of cement-based materials to a certain extent. In order to improve the shortcomings of cement-based materials and improve the application range, after a large number of experiments and studies by many scholars, it has been found that the mechanical properties and durability of composite materials have been significantly improved by adding fibers or nanomaterials to cement-based materials [3,4,5,6].

As a hollow carbon crystal curled by multilayer graphite sheets, carbon nanotubes can be divided into single-walled carbon nanotubes and multi-walled carbon nanotubes [7]. Multi-walled carbon nanotubes (MWCNTs) are the most commonly used in the field of cement-based materials. Generally, the diameter of single-walled carbon nanotubes is between 0.4–2 nm, and the diameter of multi-walled carbon nanotubes is 1.4–100 nm, and they have extremely high strength, toughness, and elastic modulus [8]. However, there are strong van der Waals forces between the tubes; therefore, carbon nanotubes are generally agglomerated. If they are directly added to the cement matrix, it is easy to cause stress concentration, which will not only fail to enhance the corresponding effect, but will also reduce the performance of the cement matrix [9,10]. Based on this consideration, researchers usually use mechanical stirring dispersion, ultrasonic dispersion, electric field-induced dispersion, surfactant modification, and other methods to achieve uniform dispersion of carbon nanotubes in the cement matrix [11,12,13].

Compared with auxiliary cementitious materials such as silica fume and fly ash, a very low content of carbon nanotube materials can achieve higher mechanical properties and durability [14,15,16,17]. For example, Cerro-Prada E et al. [18] found that only a small amount of multi-walled carbon nanotubes can increase the compressive strength and flexural strength of cement-based materials by 25% and 20%, respectively, within 90 days using conventional mechanical tests. Qin et al. [19] found that multi-walled carbon nanotubes changed the fracture process in the cement matrix and significantly increased the fracture energy when micro-cracks were initiated using laboratory tests and molecular dynamics simulations, thereby improving the overall mechanical properties. In addition, multi-walled carbon nanotubes with different sizes and morphologies have significant differences in their effects on the properties of cement-based materials. For example, Gao et al. [20] studied the effects of multi-walled carbon nanotubes with different diameters (10–20 nm, 20–40 nm and 40–60 nm) on the mechanical properties of cement-based materials. It was found that multi-walled carbon nanotubes with a diameter of 40–60 nm had the best effect on improving the flexural strength of cement, and 10–20 nm had the best effect on improving the compressive strength of cement-based materials. Ramezani et al. [21] found that the average length and average diameter of carbon nanotubes were 10–20 μm and 20–32.5 nm, respectively, which were most favorable for improving the mechanical properties of cement-based materials. Li et al. [22] used machine learning methods to predict the single-system and multi-system compressive strength of cement-based materials doped with carbon nanotubes. Some optimal parameters of carbon nanotubes were determined by the multi-system SHAP results: the optimal length and diameter of carbon nanotubes are 20 μm and 25 nm, respectively.

The improvement effect of carbon nanotubes on the mechanical properties and durability of cement-based materials is mainly attributed to its size effect and filling effect [23]. Chen et al. [24] studied the effect of carbon nanotubes on the structure of hydration products using nanoindentation tests. It was found that carbon nanotubes can bridge the pores in cement paste, promote the formation of calcium hydroxide, and promote the conversion of low-density calcium silicate hydrate to high-density calcium silicate hydrate. Wang et al. [25] used a variety of microscopic analysis methods to study multi-walled carbon nanotube cement-based materials. It was found that multi-walled carbon nanotubes can improve the hydration process, increase the number of hydration products, and reduce the porosity of cement-based materials. It has the effect of inhibiting crack propagation and improving compactness. Naqi A et al. [26] found that properly dispersed MWCNTs provide additional nucleation sites for the formation of hydrated calcium silicate (C-S-H), filling the fine pores in the cement matrix, resulting in a denser microstructure, thereby increasing strength and limiting autogenous shrinkage.

In the study of fiber-reinforced cementitious composites, steel fiber [27,28,29], basalt fiber [30,31], glass fiber [32,33], and polypropylene fiber [34,35] are often mentioned. However, steel fiber is easily agglomerated during the mixing process, and it is heavy and easy to corrode, which will incur significant cost and affect all aspects of the performance of the matrix material [36]. Polypropylene fiber has the advantages of a high tensile strength, low density, good thermal stability, and low cost [37,38]. It can significantly improve the crack resistance, toughness, and control ability of cracks caused by early shrinkage of the cement matrix [39,40,41]. For polypropylene fibers of different sizes, different contents enable them to be effectively distributed in the matrix, thereby giving the matrix a higher flexural strength [42,43,44]. The crack resistance of fiber-reinforced cement-based materials is closely related to the interface bonding between the fiber and matrix [45]. However, the application of polypropylene fiber is greatly affected by the smooth surface of polypropylene fibers, poor hydrophilicity, poor dispersion, and insufficient matrix bonding in cement mortar [46]. Researchers often use nano-coating, chemical modification, surface oxidation, etching, grafting, etc. to achieve adhesion between polypropylene fibers and matrix materials [47,48]. This method can further improve the mechanical properties and freeze–thaw resistance of composites. For example, Jia et al. [49] prepared a series of polypropylene composite fibers by melt-spinning using micro-silicon as a hydrophilic modifier and studied their structure and properties. It was found that, compared with unmodified polypropylene fibers, the composite fibers showed a rougher and more hydrophilic surface, and the interfacial bonding strength between the polypropylene fibers and cement matrix was significantly improved. Feng et al. [50] used nano-calcium carbonate to modify the surface of polypropylene fibers and incorporated them into cement-based materials. They found that the modified polypropylene fiber cement-based materials had better bending resistance. However, the above surface treatment methods for polypropylene fibers also have problems, such as a complicated operation, high equipment requirements, and poor stability.

According to the survey results, it was found that the size effect and filling effect of carbon nanotubes can effectively promote the hydration process of cement, fill holes, and make the matrix more dense, thus effectively improving the mechanical properties and durability of the cement matrix. Although polypropylene fiber can improve the flexural and tensile properties of cement-based materials, due to the lack of hydrophilic groups in its molecular chain, its bonding with the cement matrix is poor, and cracks are prone to occur at the fiber–matrix interface, thus reducing the reinforcement effects of the polypropylene fibers. Therefore, carbon nanotubes and polypropylene fibers were selected as modifiers. Multi-walled carbon nanotubes/polypropylene fiber cement-based composites were prepared via double mixing in order to demonstrate the excellent properties of multi-walled carbon nanotubes, improve the bonding performance between polypropylene fibers and the cement matrix, and achieve the purpose of improving the mechanical properties and durability of cement-based composites. The effects of multi-walled carbon nanotubes and polypropylene fibers on the mechanical properties of the cement-based materials were studied using flexural tests, compressive tests, and splitting tensile tests. The durability of the multi-walled carbon nanotubes/polypropylene fiber cement-based materials was evaluated using drying shrinkage tests and freeze–thaw cycle tests. The microstructure and pore structure of the multi-walled carbon nanotubes/polypropylene fiber cement-based materials were characterized using scanning electron microscopy and mercury intrusion tests, and the synergistic mechanisms of the multi-walled carbon nanotubes and polypropylene fibers were revealed.

## 2. Materials and Methods

### 2.1. Raw Materials

The multi-walled carbon nanotubes (MWCNTs) were produced by Suzhou Carbonfeng Technology Co., Ltd. (Suzhou, China), and their related technical indexes are shown in Table 1. The technical indexes of the polypropylene fibers (PP fibers) are shown in Table 2. The polypropylene fiber has a smooth surface, stable chemical properties, and a high aspect ratio. Its microstructure is shown in Figure 1. Polyvinylpyrrolidone (PVP) was selected as the dispersant, which is an amphiphilic polymer with a special structure. The pyrrolidone group is a hydrophilic group, and the main chain is a hydrophobic segment of the C–C bond. Introducing it into the surface of multi-walled carbon nanotubes can significantly improve their dispersion. The PVP and other raw materials and their technical indexes are shown in Table 3, which meet the requirements of the specifications.

### 2.2. Preparation of Cement-Based Composite Materials

Through many experiments in the laboratory, the following preparation process was determined. First, a certain amount of polyvinylpyrrolidone (PVP) was weighed, and the mass ratio of MWCNTs to PVP was 1:2. The amounts were added to a beaker and completely dissolved in water via magnetic stirring. Then, the MWCNTs were weighed (weighed according to the mass of the cementitious material) and added to the mixed solution via magnetic stirring. Finally, the solution was placed in the ultrasonic dispersion device, and the oscillation time was 40 min to obtain the MWCNT dispersion suspension. By consulting the relevant literature and through many experimental studies, the water-to-cementitious ratio (W/C) was determined to be 0.43. Then, polypropylene fiber (volume content) was mixed with standard sand dryly, in which the cement–sand ratio was 1:3, and cement and fly ash (fly ash replaces 20% cement quality) were added. Then, water, water reducer, and suspension were added and stirred to a uniform fluid state, and the amount of water reducer was obtained by referring to the relevant literature and debugging according to the actual mix ratio. Finally, the mixture was slowly filled into a mold that had been painted with oil, and the surface was smoothed. After being placed in an environment at 20 °C for 24 h, the mold was removed and the specimens were cured under standard curing conditions. After the curing was completed, the corresponding tests were performed, as shown in the Figure 2. The mix proportions of the cement-based composite materials are shown in Table 4.

### 2.3. Test Methods

#### 2.3.1. Fluidity Test and Mechanical Tests

The fluidity tests were carried out according to GB/T 2419-2005, and the fluidity was measured using the jump table test. According to GB/T 17671-1999, the mechanical properties of the cement-based composites were tested, including the use of flexural strength tests and compressive strength tests. The curing ages of the specimens were 3 d, 7 d, and 28 d. The flexural strength test adopted the central loading method. One side of the test body was placed on the support cylinder of the test machine. The long axis of the specimen was perpendicular to the support cylinder. A load was uniformly applied vertically to the relative side of the prism at 50 N/s through the loading cylinder until it broke. Then, the compressive strength test was carried out on the samples after the flexural test. The loading rate of the compressive strength test was 2400 N/s, and the average speed was loaded until the specimen was destroyed. The splitting tensile test was used to study the tensile properties of the cement-based materials. The specimen size was 70.7 mm × 70.7 mm × 70.7 mm. After curing for 28 days, the splitting tensile test was carried out. The qualified specimens were placed in the center of the pressure plate, the arc pads were placed above and below the pressure plate, and the loading rate was set to 0.08 MPa/s.

#### 2.3.2. Durability Tests

(1) Drying shrinkage test

The drying shrinkage test was carried out according to JC/T 603-2004. The size of the specimen was 25 mm × 25 mm × 280 mm. After curing in the standard curing box for 24 h, the mold was removed, and the specimens were placed in water at 20 °C for 48 h. After 2 days of curing, the initial length and initial weight of the specimens were tested using a specific length meter and balance. After the measurements, the specimens were placed in the curing box for drying and curing and measured again at each age (3 d, 7 d, 14 d, 21 d, 28 d, 56 d, 90 d). Finally, the dry shrinkage rate and water loss rate were calculated.

(2) Freeze–thaw cycle test

The freeze–thaw cycle test was carried out according to JGJ/T 70-2009. The size of the specimen was 40 mm × 40 mm × 160 mm. After removing the formwork, the specimens were cured in a standard curing box for 28 days, and then the specimens were immersed in water at a temperature of 20 °C for 4 days. The initial mass of the specimens was weighed, then the freeze–thaw cycle test was started. The freeze–thaw cycle test was completed in an automatic low-temperature freeze–thaw testing machine. The single freeze–thaw cycle test piece lasted 2 h, the melting lasted 0.5 h, the freezing temperature and melting temperature were −18 °C and 5 °C, respectively, and the number of freeze–thaw cycles was 100. The results of the freeze–thaw cycle test are expressed as the mass loss rate and strength loss rate. The strength test was carried out according to Section 2.3.1.

#### 2.3.3. Scanning Electron Microscopy

A MAGNA emission scanning electron microscope produced by Taisiken Co., Ltd. (Shanghai, China) was used for the test. The sample size was approximately 1 cm × 1 cm × 0.3 cm. Before the test, the selected samples were soaked in anhydrous ethanol to terminate the hydration of the cement samples. The soaking time was 24 h. After soaking, the samples were dried. Due to the poor conductivity of cement-based materials, the samples were polished and sprayed with gold. Finally, the samples were placed in the scanning chamber for observation.

#### 2.3.4. Mercury Intrusion Test

The mercury injection test was carried out using AutoPore9600/9510 automatic pore size analyzer. The maximum working pressure of the mercury injection instrument was 400 MPa, and the measurable pore diameter range was 0.003~950 μm. Firstly, after 28 days of standard curing, a long strip of 0.5–1.5 g of each of the samples was taken and immersed in anhydrous ethanol for 24 h. Then, the hydration was terminated, and the samples were dried, loaded into an expansion meter, and tested under low pressure and high pressure conditions.

## 3. Results and Analysis

### 3.1. Workability and Mechanical Properties Analysis

#### 3.1.1. Workability Analysis

The fluidity was used to evaluate the workability of the cement-based composites. The test results are shown in Table 5, and the test process is shown in Figure 3.

It can be seen from Table 5 that the addition of MWCNTs and PP fibers reduced the fluidity of the cement-based composite mixture. The contents of the MWCNTs were 0.05%, 0.1%, 0.15%, and 0.2%, and they could be decreased by 6.7%, 13.3%, 18.7%, and 20%, respectively, compared with T0. The PP fiber contents were 0.1%, 0.2%, 0.3%, and 0.4%, and the fluidity decreased by 8.0%, 14.2%, 19.6%, and 24.4%, respectively, compared with T0. When the content of PP fiber was 0.4%, the fluidity did not meet the requirements of the specifications. Compared with T0, the fluidity of T9–T12 decreased by 9.3%, 18.2%, 22.7%, and 26.7%, respectively. The fluidity of the composite materials was negatively correlated with the content of MWCNTs, and the fluidity gradually decreased with increasing contents of MWCNTs. This is due to the large specific surface area of MWCNTs. After adding the MWCNTs, the total specific surface area of the cement-based materials increased, and the free water adsorbed on the surface of the MWCNTs increased significantly. This made the cement paste become sticky, and with the increase in the MWCNT content, the fluidity of the composite material decreased more obviously. The addition of PP fiber has an adverse effect on the fluidity of the composites. When PP fiber is added to the cement-based material, a three-dimensional network structure is formed inside the mixture, which increases the friction between the aggregates. At the same time, there is a certain viscous effect between the PP fiber and the slurry, resulting in a decrease in the fluidity of the cement mixture. With the increase in fiber content, it is difficult to disperse in cement-based composites, and the distribution spacing of PP fibers becomes smaller. Therefore, the fibers were easily agglomerated during the mixing process, which hindered the fluidity of the cement paste.

#### 3.1.2. Flexural Strength and Compressive Strength

The flexural strength and compressive strength at 3 d, 7 d, and 28 d were selected to characterize the mechanical properties of the cement-based materials. The macroscopic morphology of the specimens after the flexural strength tests and compressive strength tests are shown in Figure 4. The test results are shown in Figure 5 and Figure 6.

It can be seen from Figure 5a that the incorporation of MWCNTs has a certain effect on the flexural strength of the composites. The enhancement effect increases first and then decreases with increasing contents. At the same time, the increases at 3 d and 7 d were significantly higher than that at 28 d. Compared with T0, the incorporation of MWCNTs can promote the hydration reaction of cement and have obvious improvement effects on the flexural strength of the composite materials in the early stages. When the age of the specimen was 28 d, compared with T0, the flexural strength of T1, T2, T3, and T4 increased by 5.6%, 10.1%, 15.7%, and 6.7%, respectively, and the flexural strength of T3 increased the most. It can be seen from Figure 5a that when the age of the specimen was 28 d, the flexural strength of the material increased by 10.1%, 18.0%, and 19.1%, respectively, when the PP fiber content was 0.1%, 0.2%, and 0.3% compared with T0. When the PP fiber content was 0.3%, the improvement effect was the best, and the improvement effect of the PP fiber on the flexural strength of the composite was better than that of carbon nanotubes. When the PP fiber content was 0.4%, it was reduced by 4.5% compared with the reference group. This is because the appropriate amount of PP fiber can enhance the toughness of the composite material and play a role in reinforcement and crack resistance. When the amount of PP fiber is greater, the dispersion of the fiber in the cement matrix is poor, and the stress concentration phenomenon occurs, which seriously weakens the flexural performance of the cement-based material. These results are consistent with previous research conclusions [38]. From Figure 5b, it can be seen that when the specimen age is 28 d, compared with T2 (0.1% MWCNTs), the flexural strength of the carbon nanotube/polypropylene fiber composites of T9, T10, and T11 increase by 10.2%, 18.4%, and 9.2%, respectively, while the flexural strength of the T12 scheme decreases by 10.2%. Combined with scanning electron microscopy and mercury injection tests, the incorporation of MWCNTs can improve the internal structure of PP fiber composites and reduce the porosity of the PP fiber composites. At the same time, due to the nucleation of MWCNTs [26], the hydration of cement is promoted, the bonding performance between fiber and matrix is enhanced, and the toughening effect of PP fiber is further improved. However, the excessive amount of fiber results in a decrease in the mechanical properties.

From Figure 6a, it can be seen that MWCNTs have a certain effect on the compressive strength of cement-based materials. When their contents are increased, the compressive strength shows a trend of first increasing and then decreasing. When the age of the specimen is 28 d and the MWCNT content is 0.05%, 0.1%, 0.15%, 0.2%, the compressive strength of the composite material is increased by 7.4%, 13.4%, 10.2%, and 1.3%, respectively, compared with the reference group. This is because MWCNTs have a size effect and filling effect, optimizing the pore structure of the composite material, enhancing the compactness of the structure, and giving the composite material superior compressive properties [23,24].

When the age of the specimen was 28 days, compared with the T0 group, the compressive strength of the composite was increased by 0.4%, 3.9%, and 2.4% when the PP fiber content was 0.1%, 0.2% and 0.3%, respectively. When the PP fiber content was 0.4%, the compressive strength decreased by 9.7% compared with the T0 group. It was found that when the content of PP fiber was too low, the fibers could not form a strong support system, having little effect on the compressive strength. When the fiber content was too large, the fibers agglomerated in the composite material and could not inhibit the development of small cracks. At the same time, the fiber agglomeration increased the number of internal pores and reduced the density of the composite material, thus causing the compressive strength of the composite material to decrease. These findings are consistent with the results of Li [39] and Al-Katib [41]. It can be seen from Figure 6b that when the age of the specimen is 28 days, compared with the T2 group, the compressive strength of the carbon nanotube/polypropylene fiber composites of the T9 and T10 schemes is increased by 8.5% and 9.3%, respectively. When the content of PP fiber in the composite group is 0.3% and 0.4%, it is reduced by 0.4% and 12.8%, respectively. This shows that, although the combined addition of MWCNTs and PP fiber can further improve the compressive strength of the composite, the enhancement effect cannot be completely superimposed. When the content is too high, the compressive strength of the composite is weakened.

#### 3.1.3. Split Tensile Strength

The splitting tensile strength at 28 d was used to characterize the tensile stress resistance of the cement-based materials. The macroscopic morphology of the specimens after the split tensile strength test is shown in Figure 7. The splitting tensile test results are shown in Figure 8.

It can be seen from Figure 8 that when the content of MWCNTs is 0.05%, 0.1%, 0.15%, and 0.2%, the splitting tensile strength of the cement-based composites increases by 2.9%, 6.8%, 10.7%, and 4.2%, respectively, compared with the reference group. MWCNTs have a nanoscale size and excellent tensile strength, which have good filling effects on the internal micro-pores of the composite material. They also improve the compactness of the matrix and play a bridging role to a certain extent, which has a certain inhibitory effect on the development of cracks [19,24]. When the content of PP fiber was 0.1%, 0.2%, and 0.3%, the splitting tensile strength of the cement-based composites increased by 1.9%, 9.1%, and 12.6%, respectively, compared with the reference group, while it was decreased by 1.6% when the fiber content was 0.4%. The toughening and crack resistance effects of PP fiber have a significant effect on the splitting tensile strength of the composite material. These results are consistent with previous studies [40,44]. When the matrix reaches the critical failure load, due to the existence of the fiber, the concentrated stress at the crack is transferred to the other interface of the matrix through the fiber, which reduces the stress concentration at the crack end and inhibits the continuous development of the crack. Compared with 0.1% MWCNTs, the splitting tensile strength of the 0.1%, 0.2%, and 0.3% composites can be increased by 4.2%, 6.7%, and 5.8%, respectively. When the PP fiber content is 0.4%, it can be decreased by 6.7%. Compared with the single-doped method, the composite-doped method shows more excellent tensile strength. Combined with the results of the scanning electron microscopy and mercury intrusion tests, it can be assumed that the addition of MWCNTs improves the bonding degree of the PP fiber–cement interface and ensures the toughening and crack resistance effects of the PP fiber on composite materials.

### 3.2. Durability Analysis

#### 3.2.1. Drying Shrinkage Test

The durability of the cement-based composites was characterized by the dry shrinkage and water loss rate. The test results are shown in Figure 9.

From Figure 9a, it can be seen that when the MWCNT content was 0.1%, the reduction in the drying shrinkage rate of the composite material was the best. With increasing curing ages, it showed a trend of first increasing and then decreasing. Compared with the reference group, the drying shrinkage rate of the composite with 0.1% MWCNT was decreased by 15.7%, 16.1%, 13.5%, 13.1%, 12.5%, and 12.1% at 3 d, 7 d, 14 d, 28 d, 60 d, and 90 d, respectively. The incorporation of MWCNTs significantly improves the drying shrinkage of the composite. This is because, on the one hand, the size effect and filling effect of MWCNTs significantly improve the pore structure of the composite material. At the same time, when 0.1%MWCNTs is added to the cement-based material, the water loss rate of the composite material improves the most. The water loss rate showed a trend of decreasing first and then increasing with increasing MWCNT contents, and the improvements in the water loss rate gradually decreased with increasing age. When the content of the MWCNTs is too much, it agglomerates locally in the matrix, destroys the capillary pore structure of the composite, and accelerates the water loss. From Figure 9b, it can be seen that when the content of PP fiber is 0.3%, the drying shrinkage rate of the composite material reduces the most. The drying shrinkage rate of the composite material with 0.3% PP fiber at the ages of 3 d, 7 d, 14 d, 28 d, 60 d, and 90 d was reduced by 13.3%, 13.8%, 14.5%, 16.4%, 14.4%, and 14.3%, respectively, compared with the T0 group. In addition, the PP fiber reduced the water loss rate of the composites. At the early ages, with the increase in curing age, the reduction effect is more significant. When the curing age exceeded 28 d, the water loss rate of the composite material decreased, and the water inside the composite material was lost. When the content of PP fiber was 0.3%, the water loss rate of the composite was the lowest. When the fiber content is low, the fiber is distributed inside the cement paste, and the fiber is almost not observed on the surface of the matrix. This phenomenon results in an increase in the water loss channels generated by the fiber and the slurry, resulting in an increase in the water loss rate. With the increasing fiber contents, the water loss rate gradually decreases. This is because the fiber is not only dispersed inside, but it is also distributed on the surface of the matrix. Because the water loss channel is blocked by the fiber, the water loss path inside the composite is reduced, and the water evaporation rate is reduced. It can be seen in Figure 9c that the dry shrinkage rates of the cement-based composites with carbon nanotubes/PP fibers at 3 d, 7 d, 14 d, 28 d, 60 d, and 90 d were 4.3%, 8.7%, 2.1%, 7.0%, 5.0%, and 4.6% lower than those of the 0.1% MWCNT group, respectively. When the MWCNTs and PP fibers act simultaneously in the composite, the drying shrinkage of the composite is further improved, and the two materials jointly bear the anti-drying shrinkage ability of the composite. Using the compound blending combination method, when the PP fiber content was 0.2%, the water loss rate of the composite material was further reduced, and the drying shrinkage improvement effect was also optimal.

#### 3.2.2. Freeze–Thaw Cycle Test

The mass loss rate and strength loss rate were used to characterize the durability of the cement-based materials. The macroscopic morphology of the specimen after the test is shown in Figure 10, and the test results are shown in Table 6 and Figure 11.

It can be seen from Table 6 that, except for the T0 group, the mass loss rates of the other 12 groups of specimens were negative after 25 freeze–thaw cycles. The reason for this phenomenon may be due to the water absorption of the carbon nanotubes and PP fibers. When the freeze–thaw cycle was completed 50 times, the cement was fully hydrated, the adsorbed water inside the material was saturated, and the mass loss rate began to be greater than the mass growth rate due to freeze–thaw damage. As the freeze–thaw test continued, the degree of damage to the specimen gradually increased, and the mass loss rate increased sharply. The MWCNTs were added to the cement-based materials and subjected to 100 freeze–thaw cycles. The mass loss rate of the composite material can be reduced to a certain extent compared with the reference group, and the reduction is different for different dosages. When the MWCNT contents were 0.05%, 0.1%, 0.15%, and 0.2%, the mass loss rate was reduced by 17.2%, 27.9%, 23.1%, and 15.2%, respectively, compared with the reference group. It can be seen from the mass loss rate of the composite material that the optimal content of the MWCNTs was 0.1%. According to the results of the scanning electron microscopy and mercury intrusion tests, it can be seen that MWCNTs can optimize the pore structure, enhance the density of the material, and improve the freeze–thaw resistance. In addition, when the composite material produces expansion pressure during the freeze–thaw cycle, the bridging effect and excellent mechanical properties of the MWCNTs can resist part of the pressure and inhibit the generation of cracks. When the content of the PP fiber was 0.1%, 0.2%, 0.3%, and 0.4%, the mass loss rate was 14.3%, 25.2%, 17.2%, and 10.4% lower than that of the reference group, respectively. It can be seen from the mass loss of the composite material that the optimum content of PP fiber is 0.2%. During the freeze–thaw cycle, when the cement matrix is subjected to expansion pressure, cracks begin to appear inside the matrix, and the bridging effect of the fiber inhibits the expansion of the cracks to a certain extent. At the same time, due to the binding effects of the PP fiber, the integrity of the composite material is ensured, and the anti-stripping ability is improved. In the composite group (T9–T12), when the content of PP fiber was 0.1%, 0.2%, 0.3%, and 0.4%, the mass loss rate was reduced by 22.4%, 34.9%, 26.3%, and 14.7%, respectively, after 100 freeze–thaw cycles compared with the reference group. When the content of PP fiber was 0.2%, the improvement effect on the mass loss of the composite material was the best.

It can be seen from Figure 11 that the strength loss of composite materials in all the test schemes increased with increasing numbers of freeze–thaw cycles. This is because with increasing numbers of freeze–thaw cycles, the matrix continues to be subjected to freeze–thaw damage, the internal micro-cracks continue to expand, and the compactness decreases, resulting in a sharp decline in the mechanical properties. When the number of freeze–thaw cycles is 50 and the MWCNT content is 0%, 0.05%, 0.1%, 0.15%, and 0.2%, the flexural strength loss rate of the composite is 40.2%, 31.2%, 30.7%, 29.2%, and 36.5%, respectively, and the compressive strength loss rate of the composite is 27.0%, 18.3%, 17.0%, 17.7%, and 19.7%, respectively. When the number of freeze–thaw cycles reaches 100 and the MWCNT content is 0%, 0.05%, 0.1%, 0.15%, and 0.2%, the loss rates of the flexural strength of the composites are 82.8%, 73.1%, 69.3%, 73.6%, and 77.1%, respectively, and the loss rates of the compressive strength of the composites are 61.9%, 55.4%, 52.8%, 54.5%, and 56.5%, respectively. It can be found that the addition of MWCNTs significantly improved the strength loss of the composites under the freeze–thaw cycles. When the number of freeze–thaw cycles is 50 and the content of PP fiber is 0%, 0.1%, 0.2%, 0.3%, and 0.4%, the flexural strength loss rates of the composites are 40.2%, 33.0%, 27.5%, 34.6%, and 43.8%, respectively, and the compressive strength loss rates of the composites are 27.0%, 19.8%, 18.6%, 21.0%, and 27.9%, respectively. When the number of freeze–thaw cycles reaches 100 and the PP fiber content is 0%, 0.1%, 0.2%, 0.3%, and 0.4%, the flexural strength loss rate of the composite is 82.8%, 77.7%, 68.6%, 72.9%, and 84.3%, respectively, and the compressive strength loss rate of the composite is 61.9%, 58.2%, 57.9%, 58.4%, and 61.9%, respectively.

The incorporation of PP fiber has a significant improvement in the toughness of the cement-based materials. At the same time, the three-dimensional network structure formed by the fiber inside the composite material improves the overall stability of the composite material. Therefore, it has a higher resistance to freeze–thaw cycles than the reference group, and the strength loss is also reduced. When the number of freeze–thaw cycles was 50, the flexural strength loss rates of the T9–T12 cement-based composites were 31.5%, 28.4%, 33.3%, and 36.7%, respectively, and the compressive strength loss rates of the composites were 18.0%, 16.8%, 19.5%, and 22.0%, respectively. When the number of freeze–thaw cycles reached 100, the flexural strength loss rates of the T9–T12 cement-based composites were 70.4%, 66.4%, 70.6%, and 77.8%, respectively, and the compressive strength of the composites were 55.2%, 52.2%, 55.4%, and 57.6%, respectively. Compared with the single-doped combination method, the composite-doped method enables the cement matrix to maintain higher mechanical properties under freeze–thaw cycles.

### 3.3. Scanning Electron Microscopy Test Analysis

The microstructures of T0, T2, T6, and T10 were observed using scanning electron microscopy. The test results are shown in Figure 12, Figure 13, Figure 14 and Figure 15.

By comparing Figure 12 and Figure 13, it can be seen that a large number of hydration products were attached to the surface of the composites doped with MWCNTs, and the number of tiny pores was significantly reduced. This shows that MWCNTs have a significant filling effect on the micropores of the composite material, exerting an inhibitory effect on the generation of micro cracks and making the composite material more dense. Figure 13c shows the micromorphology of the crack when it is magnified 10,000 times. It can be found that some MWCNTs are lapped at both ends of the crack, and some MWCNTs are pulled off. When the crack appears and further expands, the presence of MWCNTs delays the continued expansion of the crack and transfers the stress of the crack propagation to different spaces inside the material.

From Figure 14a, it can be seen that some fibers are in a state of tensile fracture. When the fiber is subjected to tensile stress, tensile deformation occurs. When it exceeds its own tensile strength, the fiber is broken. The PP fiber reduces the brittleness of the composite material through the bridging effect. Figure 14b shows the morphology of the joint of PP fiber and cement matrix under 5000 times magnification. It can be seen that PP fiber plays a certain role in hindering the expansion of cracks. When the crack appears and begins to develop, the crack propagation is hindered due to the presence of the fiber. When the crack changes the development direction or passes through the fiber, the internal stress field of the composite material is weakened to a certain extent, and the concentrated stress at the crack tip makes it difficult to support the crack propagation. At the same time, it can be seen that there are small cracks in the interface transition zone between the fiber and the cement matrix. However, there are some hydration products around the fiber, indicating that the combination of PP fiber and cement paste has certain deficiencies, and the toughening effect of the composite material cannot be fully reflected. From Figure 14c, it can be found that a certain amount of C-S-H is attached to the surface of the fiber. These hydration products have a strong bonding force with each other, and the fiber and the matrix are anchored together. The more hydration products, the better the anchoring effect.

By comparing Figure 15 and Figure 14, it can be seen that there is a better bonding effect between the fiber and the matrix. Figure 15b shows a microscopic diagram of the connection between the PP fiber and cement matrix under 10,000 times magnification. It can be seen that the multi-walled carbon nanotube/polypropylene fiber cement-based material has more hydration products than the polypropylene fiber cement-based material. This is due to the fact that PP fiber has no obvious improvement effects on cement hydration, while the addition of MWCNTs promotes an increase in hydration products. The distribution of C-S-H gel is more dense, and a large number of hydration products wrap the PP fiber, which strengthens the bonding ability between the fiber and matrix. At the same time, MWCNTs can fill the tiny pores generated when PP fibers are combined with the matrix, enhancing the compactness of the internal structure, improving the binding capacity of PP fiber and cement matrix, and enhancing the toughening effect of PPA.

### 3.4. Mercury Intrusion Test

Mercury injection experiments were carried out on the T0, T2, T6, and T10 specimens, respectively, to study the influence of multi-walled carbon nanotubes and polypropylene fibers on the pore structure of the cement-based materials. The experimental results are shown in Table 7 and Figure 16. The internal pores of the composite material were divided into four categories [51]: harmless pores less than 20 nm, less harmful pores between 20 nm and 50 nm, harmful pores between 50 nm and 200 nm, and more harmful pores greater than 200 nm. The porosity of the different types of pores was counted, and the pore size distribution is shown in Table 8.

It can be seen from Figure 16 that the porosity curve of the T2 group was lower than that of the T0 group, and it gradually tended to 0 with increasing pore sizes. At the same time, according to the Table 7, the pore structure parameters, such as the average pore size and median pore size, of the T2 group were lower than those of the T0 group. This indicated that the multi-walled carbon nanotubes could effectively change the pore structure of the cement-based materials. By comparing T0 and T6, it can be seen that the addition of PP fibers had an adverse effect on the overall pore structure of the composite. However, compared with the T6 group, the pore structure parameters of the T10 group were improved to some extent, indicating that the incorporation of MWCNTs had a certain refinement effect on the pore size of the PP fiber composite. This is due to the filling effect of MWCNTs and the promotion of cement hydration. The improvement in the pore structure improves the compactness of the composite. It can be seen from Table 8 that the number of harmful pores and multi-harmful pores of the composites was reduced through the optimization and improvements in the pore structures of MWCNTs, and the proportion of harmless pores and less harmful pores was increased. Since PP fiber may bring in trace air when it is added to cement-based materials, bubbles are generated in the composite material, resulting in an increase in harmful pores. At the same time, due to the insufficient bonding ability of the fiber in the composite material with the cement slurry, the porosity of the composite material increases after the fiber is added. The number of harmful pores and multi-harmful pores in the T10 group was lower than that in the T6 group. Therefore, the addition of MWCNTs in the PP fiber composite improved the pore structure to a certain extent, reduced the adverse effects of PP fibers on the internal structure of the composite, and enhanced the compactness of the composite.

## 4. Conclusions

(1) When the content of MWCNTs is 0.15%, the 28 d flexural strength and splitting tensile strength of the composites are the best, which are 15.7% and 10.7% higher than those of the reference group, respectively. Compared with the reference group, when the content of MWCNTs is 0.1%, the compressive strength of the composite is improved by 9.5%. At the same time, the durability of the composite is also significantly improved. The 90 d dry shrinkage rate and water loss rate are reduced by 12.1% and 4.2%, respectively. After 100 freeze–thaw cycles, the mass loss rate, flexural strength loss rate, and compressive strength loss rate are reduced by 27.9%, 16.3%, and 14.7%, respectively.

(2) When the content of PP fiber is 0.3%, compared with the reference group, the flexural strength and splitting tensile strength of the composites at 28 d can be increased by 19.1% and 12.6%, respectively, and the dry shrinkage rate and water loss rate at 90 d can be decreased by 14.3% and 6.1%, respectively. When the content of PP fiber is 0.2%, the 28 d compressive strength can be increased by 3.9% compared with the reference group. After 100 freeze–thaw cycles, the mass loss rate, compressive strength loss rate, and flexural strength loss rate can be decreased by 25.2%, 17.1%, and 6.5%, respectively, compared with the reference group.

(3) The mechanical properties and durability of the 0.1% MWCNTs/0.2% PP fiber composites are further improved. At the age of 28 d, the flexural strength, compressive strength, and splitting tensile strength can be increased by 18.4%, 9.3%, and 6.7%, respectively. The dry shrinkage rate and water loss rate of the composites at 90 d are 4.6% and 3.3% lower than those of the single-doped MWCNTs, respectively. After 100 freeze–thaw cycles, the mass loss rate, flexural strength loss rate, and compressive strength loss rate are reduced by 9.7%, 4.2%, and 1.1%, respectively, compared with the single-doped MWCNTs.

(4) MWCNTs significantly improve the internal structure of the composite material, promote the production of cement hydration products, and improve the compactness of the composite material. The porosity of the single-doped MWCNTs is 14.8% lower than that of the T0 group. PP fiber has obvious toughening and crack resistance effects on the composites. However, the porosity of the single-doped PP fiber is 0.7% higher than that of the T0 group. When the MWCNTs and PP fibers are mixed, the porosity of the composite material is 10.0% lower than that of the single-doped PP fiber. This shows that adding MWCNTs to the PP fiber composite material can make up for the adverse effects of the PP fiber, enhance the bonding ability between the fiber and the cement matrix, and improve the compactness of the composite material.

## Figures and Tables

**Figure 1 polymers-16-00041-f001:**
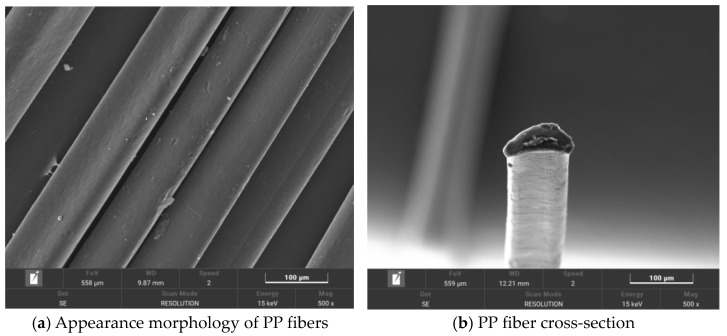
SEM of PP fibers.

**Figure 2 polymers-16-00041-f002:**
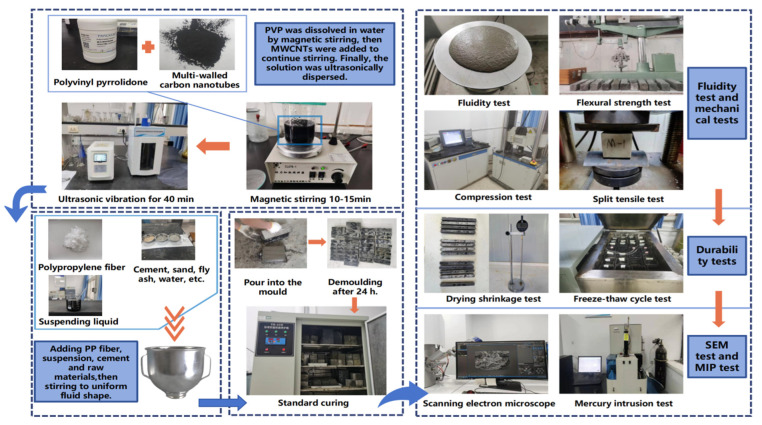
Preparation process and test flow chart.

**Figure 3 polymers-16-00041-f003:**
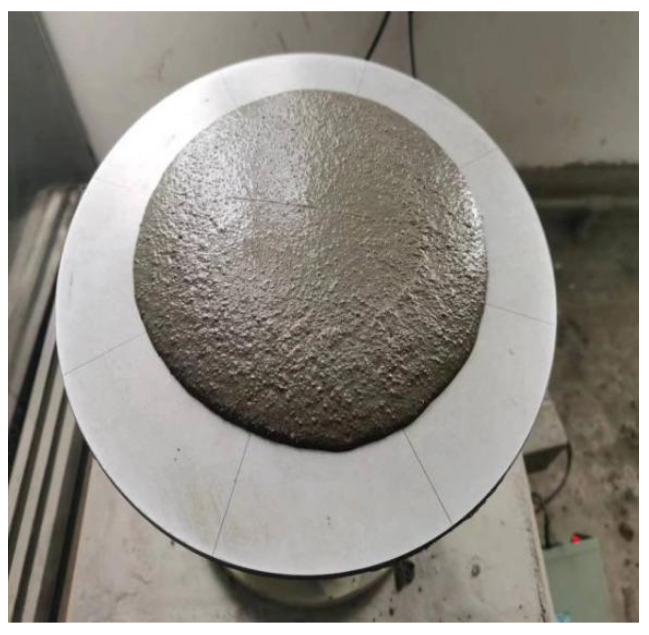
Fluidity test.

**Figure 4 polymers-16-00041-f004:**
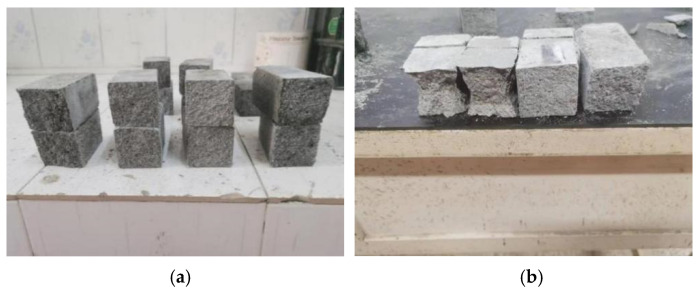
The macroscopic morphology of the specimens. (**a**) The macroscopic morphology of the specimens after the flexural strength test. (**b**) The macroscopic morphology of the specimens after the compressive strength test.

**Figure 5 polymers-16-00041-f005:**
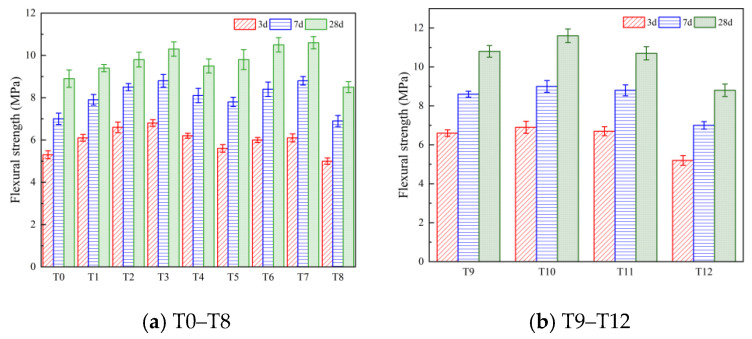
Flexural strength test results.

**Figure 6 polymers-16-00041-f006:**
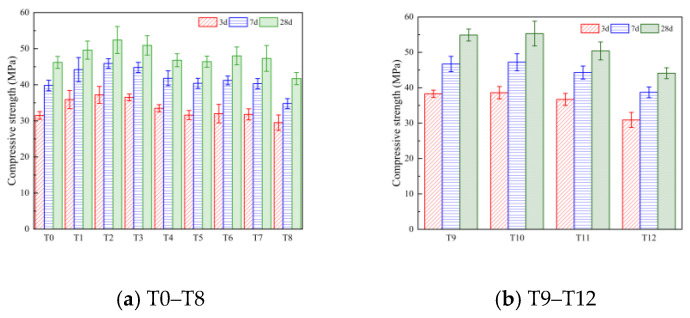
Compressive strength test results.

**Figure 7 polymers-16-00041-f007:**
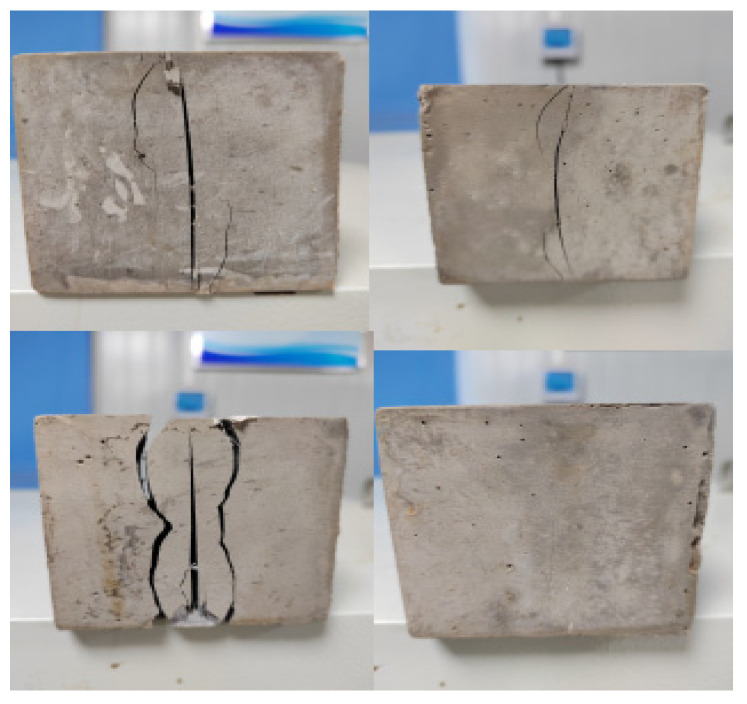
The macroscopic morphology of the specimens after the split tensile strength test.

**Figure 8 polymers-16-00041-f008:**
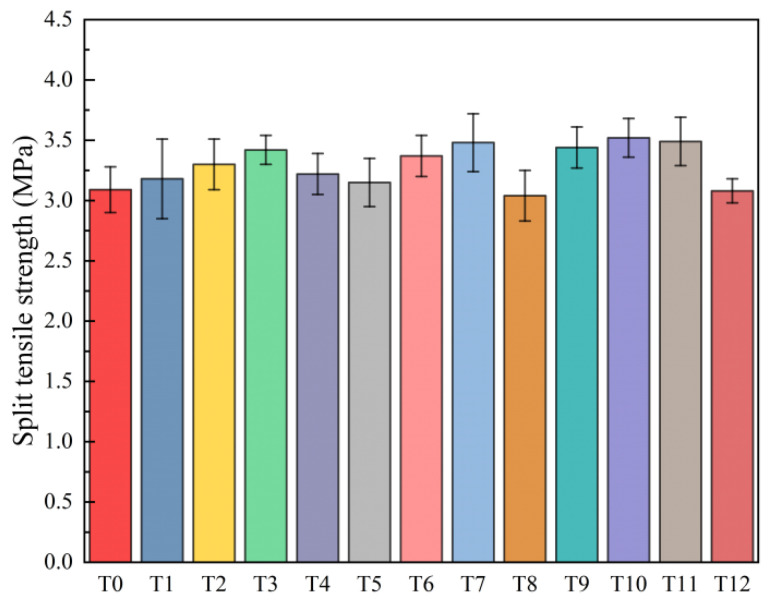
T0–T12 results of the splitting tensile tests.

**Figure 9 polymers-16-00041-f009:**
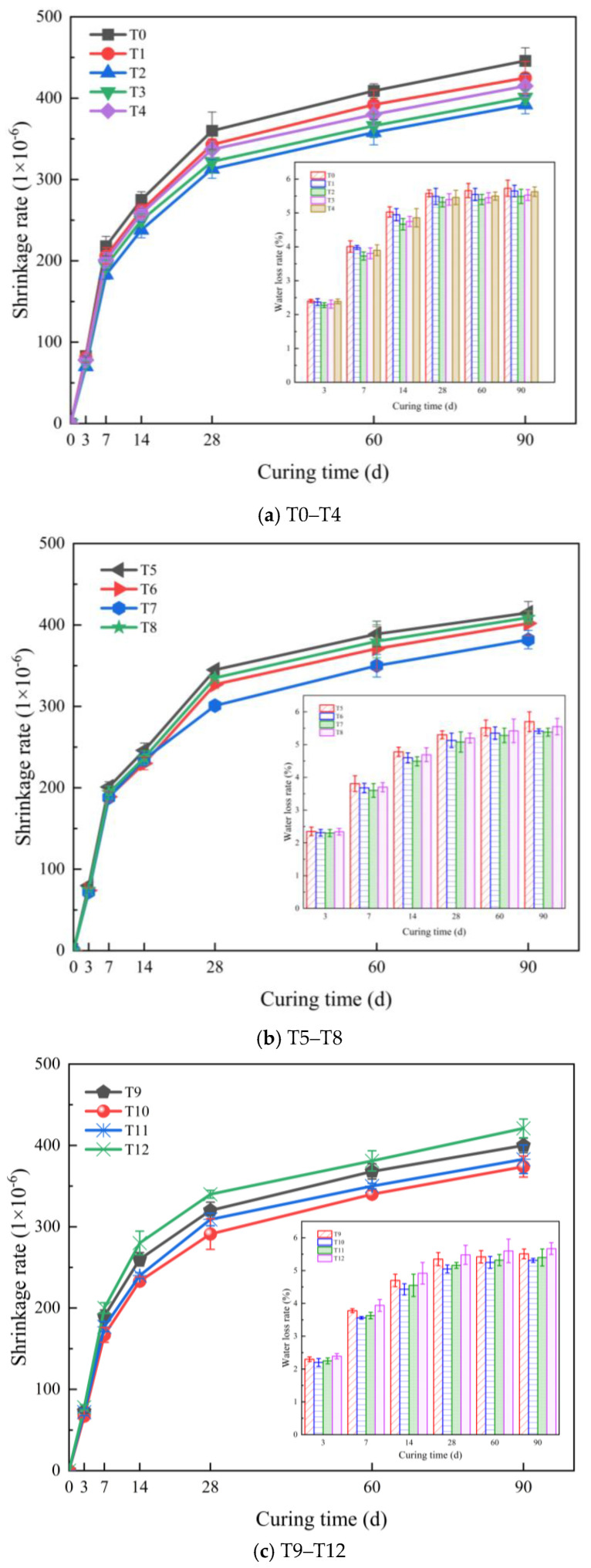
Results of the drying shrinkage tests.

**Figure 10 polymers-16-00041-f010:**
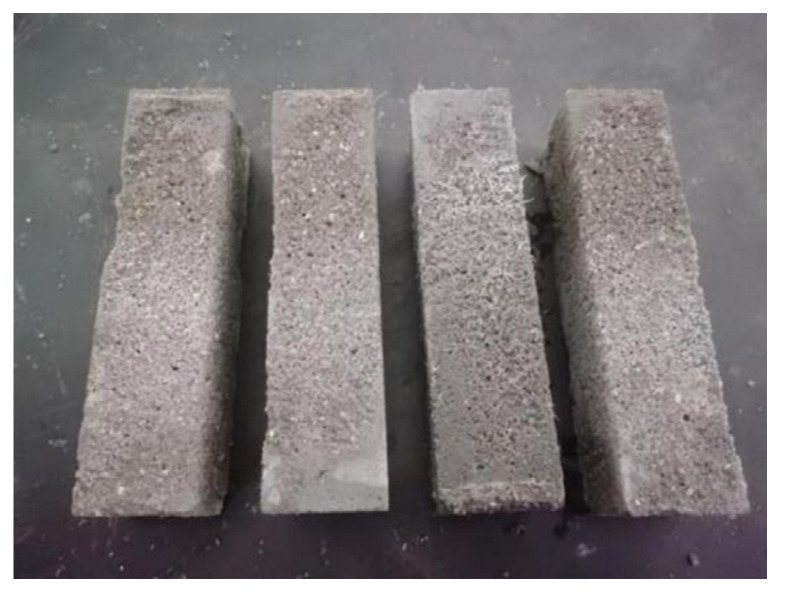
The macroscopic morphology of the specimens after the freeze–thaw cycle test.

**Figure 11 polymers-16-00041-f011:**
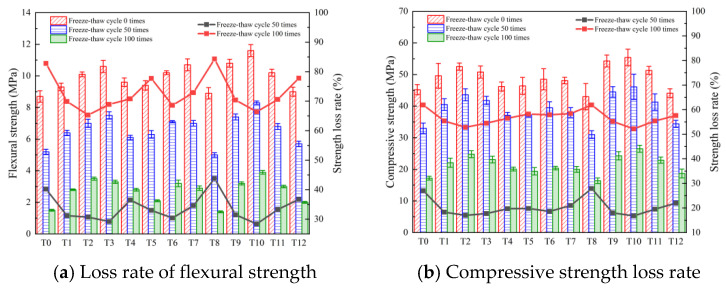
Strength loss rate.

**Figure 12 polymers-16-00041-f012:**
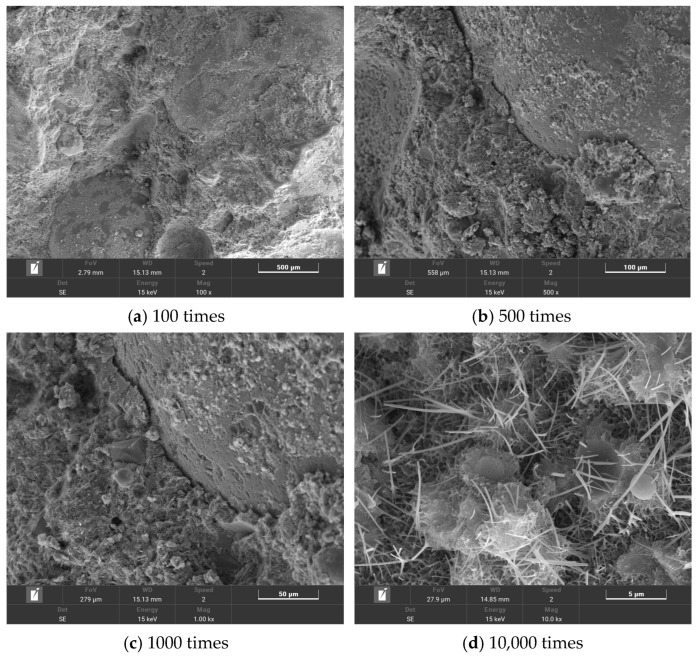
T0 microtopography.

**Figure 13 polymers-16-00041-f013:**
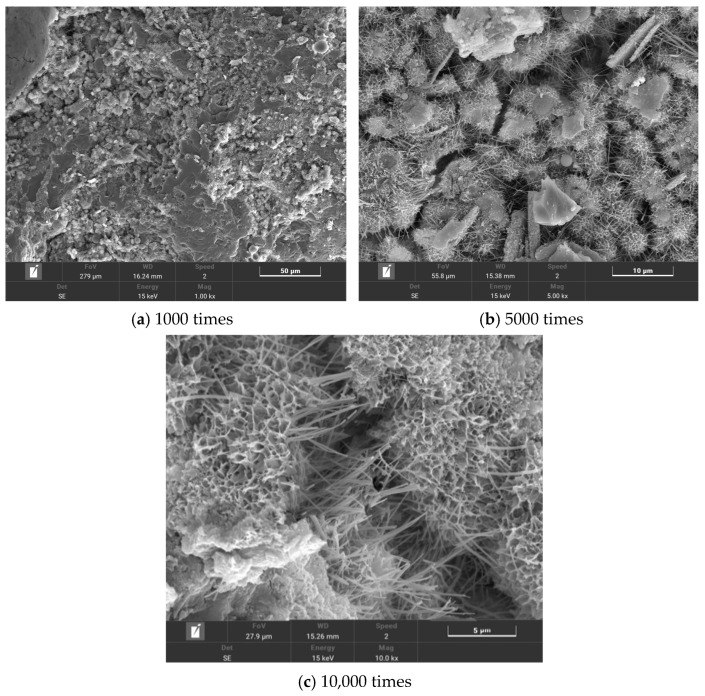
T2 microtopography.

**Figure 14 polymers-16-00041-f014:**
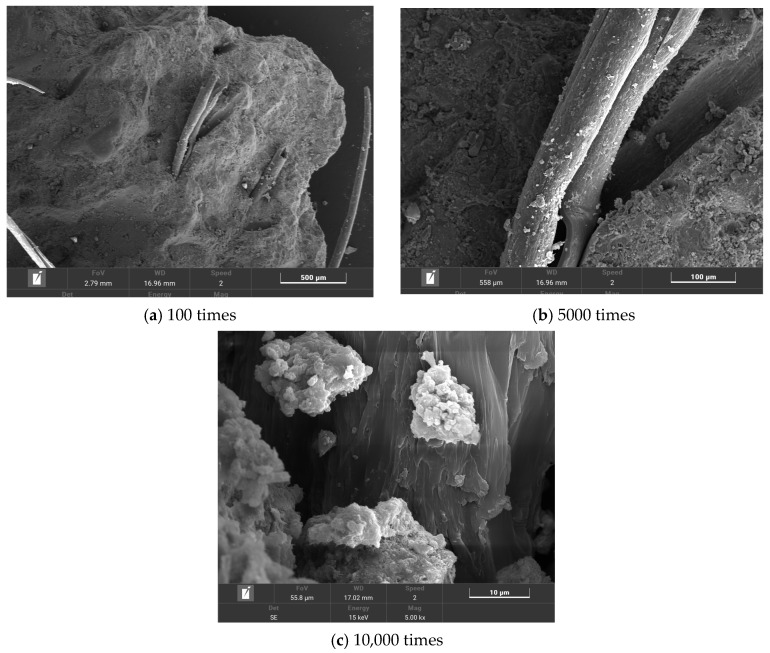
T6 microtopography.

**Figure 15 polymers-16-00041-f015:**
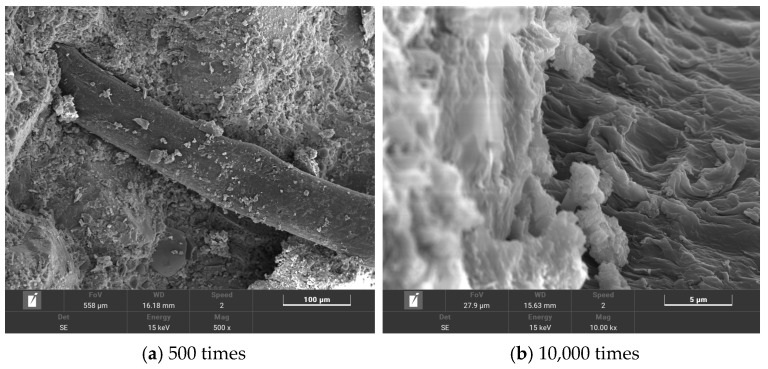
T10 microtopography.

**Figure 16 polymers-16-00041-f016:**
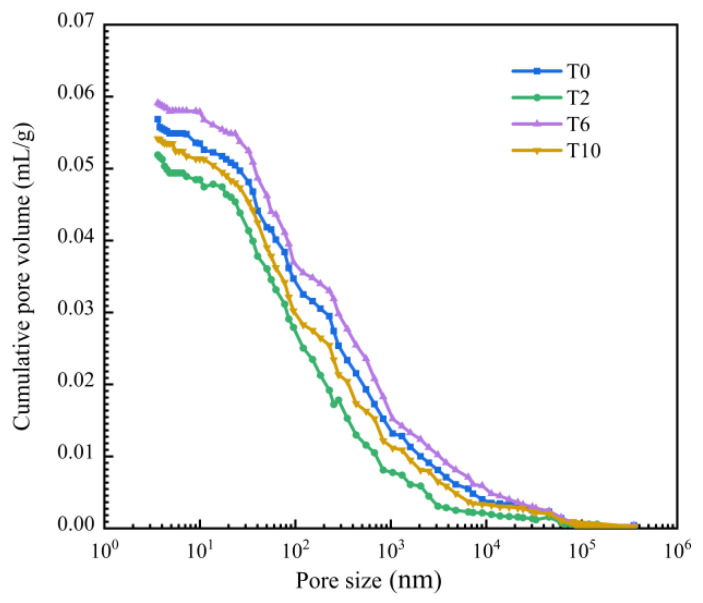
Cumulative pore volume.

**Table 1 polymers-16-00041-t001:** Technical indexes of multi-walled carbon nanotubes.

Item	Diameter (nm)	Length (μm)	Purity (wt%)	Specific Surface Area (%)
MWCNTs	10–20	5–15	>97%	90–120 m^2^/g

**Table 2 polymers-16-00041-t002:** Technical indexes of polypropylene fibers.

Item	Indexes
Fiber type	Bunchy monofilament
Tensile strength (MPa)	>486
Elastic modulus (GPa)	>4.8
Melting point (°C)	169
Density (g/cm^3^)	0.91
Length (mm)	9
Diameter (μm)	18–48

**Table 3 polymers-16-00041-t003:** Related raw materials and indexes.

No.	Types	Materials	Indexes
1	Binding material	P.O 42.5 Cement	Fineness, 5.4%; standard consistency water consumption, 26.1%; initial setting time, 255 min; final setting time, 365 min; soundness, 1.2 mm
2	Binding material	Fly ash (I grade)	Mean diameter, 20.13 μm; fineness, 9.8%; water demand ratio, 93.1%; moisture content, 0.2%; loss on ignition, 1.35%
3	Dispersant	Polyvinyl pyrrolidone	White powder; K-value, 27.0–32.4; pH value, 3.0–5.0; total nitrogen content, 11.5–12.8%; ignition residue, ≤0.1%; aldehyde content, ≤0.05%; formic acid content, ≤0.5%; vinylpyrrolidone content, ≤0.001%; plumbum content, ≤0.001%; water content ≤ 5.0%
4	Auxiliary material	Naphthalene water reducer	Yellow–brown powder; water reduction, 8–14%; bleeding rate, 55%; gas content, 3.0%; 28 d shrinkage ratio, 110%
5	Sand	ISO standard sand	Grain diameter, 0.08–2 mm

**Table 4 polymers-16-00041-t004:** Mix proportions of cement-based composite materials.

Scheme	W/C	MWCNTs (wt%)	PP Fiber (%)
T0	0.43	0	0
T1	0.43	0.05	0
T2	0.43	0.1	0
T3	0.43	0.15	0
T4	0.43	0.2	0
T5	0.43	0	0.1
T6	0.43	0	0.2
T7	0.43	0	0.3
T8	0.43	0	0.4
T9	0.43	0.1	0.1
T10	0.43	0.1	0.2
T11	0.43	0.1	0.3
T12	0.43	0.1	0.4

**Table 5 polymers-16-00041-t005:** Fluidity of the cement-based composites.

Scheme	T0	T1	T2	T3	T4	T5	T6	T7	T8	T9	T10	T11	T12
Fluidity (mm)	225	210	195	183	180	207	193	181	170	204	184	174	165

**Table 6 polymers-16-00041-t006:** The mass loss rate test results.

Scheme	Freeze–Thaw Cycle (Cycles)			
	25	50	75	100
T0	0.11	0.85	2.45	4.41
T1	−0.08	0.30	2.10	3.65
T2	−0.23	0.19	1.90	3.18
T3	−0.18	0.26	2.01	3.39
T4	−0.14	0.46	2.20	3.74
T5	−0.02	0.37	2.11	3.78
T6	−0.05	0.21	1.95	3.30
T7	−0.03	0.30	2.02	3.65
T8	−0.01	0.55	2.23	3.95
T9	−0.10	0.41	2.10	3.42
T10	−0.25	0.14	1.78	2.87
T11	−0.13	0.36	2.01	3.25
T12	−0.05	0.54	2.28	3.76

**Table 7 polymers-16-00041-t007:** Mercury injection test results.

Scheme	Average Pore Size (nm)	Medium Pore Diameter (nm)	Median Volume (cc g^−1^)	Median Surface Area (m^2^ g^−1^)
T0	73.31	128.23	0.0287	2.253
T2	59.95	91.33	0.0175	1.452
T6	87.94	135.21	0.0301	2.295
T10	69.82	108.45	0.0195	1.784

**Table 8 polymers-16-00041-t008:** Pore size distribution test results.

Scheme	Pore Size Distribution (%)				Most Probable Pore Size (nm)	Porosity (%)
	<20 nm	20–50 nm	50–200 nm	>200 nm		
T0	2.3183	3.0358	1.2712	2.1205	40.2718	8.7458
T2	2.3472	3.0868	0.8142	1.2002	26.2991	7.4484
T6	2.2501	2.8872	1.4742	2.1928	45.7543	8.8043
T10	2.2748	2.9015	0.9177	1.8334	36.3927	7.9274

## Data Availability

Some or all of the data, models, or code that support the findings of this study are available from the corresponding author upon reasonable request.

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
