# Peer review of "Study on the Properties of Multi-Walled Carbon Nanotubes (MWCNTs)/Polypropylene Fiber (PP Fiber) Cement-Based Materials"

_polymers, 2023, doi:10.3390/polym16010041_

Round 1

Reviewer 1 Report

Comments and Suggestions for Authors

The authors conducted some experimental tests to determine cementitious composites' mechanical properties and durability characteristics containing multi-walled carbon nanotubes (MWCNTs) and polypropylene. Although the results interest readers, a major revision is required to improve the manuscript. Also, the reviewer is concerned about the novelty of the work, as many studies reported the effect of nanomaterials within mortar and cement paste. Please find the following comments:

-          Abstract: Please revise the first sentence of the abstract.

-          Please also mention nano coating on the surface of fibres to solve the issue of fibres’ surface. The authors can use the following reference to explain this method.

“Ahmadi, K., Mousavi, S. S., & Dehestani, M. (2023). Influence of nano-coated micro steel fibers on mechanical and self-healing properties of 3D printable concrete using graphene oxide and polyvinyl alcohol. Journal of Adhesion Science and Technology, 1-22.”

-          The authors need to explain “Carbon nanotubes/polypropylene fibre cement-based composites were prepared by compounding.”.

-          Please accurately list the objectives of the present study at the end of the introduction section.

-          Please explain the reason for using PVP in this study.

-          General comments: Please do not use short paragraphs throughout the manuscript.

-          Table 4: The W/C is water-to-cement or water-to-cementitious ratio? Why 0.43 was selected for this study?

-          Using nanomaterials reduces the workability of mortar along with using fibres. How did the authors consider this point? The mini-slump and mixtures' workability should be constant when comparing the mechanical characteristics. The reviewer recommends reporting mini-slump reports in Table 2 and explaining how much SP was used for each mixture.

-          Table 2: percentages of MWCNTs are cement replacement (mass), but fibres should be volume fraction percentages. Please mention this within the table.

-          The authors must explain why the mixtures T9 to T12 were designed. Changing PP and MWCNTs percentages cannot be properly analysed without statistical analysis.

-          Please improve the quality of Figures 3-5.

-          General comment: improving the compressive strength of cement-based composites containing nanomaterials was severely reported by researchers; accordingly, there is some concern regarding the novelty of the study. Please explain.

-          Please add some photos of samples before and after performing mechanical tests.

-          The reviewer recommends analysing the database of mechanical tests using statistical software for ANOVA to determine the synergistic effects of nanomaterial and fiber.

-          Please enhance the quality of Figures 6-7 & 12.

-       Section of 3.2.2. Freeze-thaw cycle test: Please add some photos of samples before and after F/T cycles.

Comments on the Quality of English Language

Minor editing of English language required.

Author Response

Dear Reviewer:

Thank you for your valuable comments, we have carried on the careful discussion and modification, the specific reply is as follows:

1.Abstract: Please revise the first sentence of the abstract.

Reply:Thank you for your valuable comments. We have modified the abstract part and emphasized it with yellow.

2.Please also mention nano coating on the surface of fibres to solve the issue of fibres’ surface. The authors can use the following reference to explain this method.

“Ahmadi, K., Mousavi, S. S., & Dehestani, M. (2023). Influence of nano-coated micro steel fibers on mechanical and self-healing properties of 3D printable concrete using graphene oxide and polyvinyl alcohol. Journal of Adhesion Science and Technology, 1-22.”

Reply:Thank you for your valuable comments. We have supplemented in the Introduction and References, and emphasized with yellow.

3.The authors need to explain “Carbon nanotubes/polypropylene fibre cement-based composites were prepared by compounding.”.

Reply:Thank you for your valuable comments. We apologize for the previous statement. We have modified it and highlighted it in yellow. As follows : Multi-walled carbon nanotubes/polypropylene fiber cement-based composites were prepared by double mixing.

4.Please accurately list the objectives of the present study at the end of the introduction section.

Reply:Thank you for your valuable comments. We have supplemented it at the end of the introduction and emphasized it in yellow.

  1. Please explain the reason for using PVP in this study.

Reply:Thank you for your valuable comments. Polyvinylpyrrolidone (PVP) was selected as the dispersant, it was an amphiphilic polymer with a special structure. The pyrrolidone group is a hydrophilic group, and the main chain is a hydrophobic segment of the C−C bond. Introducing it into the surface of multi-walled carbon nanotubes can significantly improve the dispersion of multi-walled carbon nanotubes. We have supplemented the role of PVP in Section 2.1 and emphasized it with yellow.

6.General comments: Please do not use short paragraphs throughout the manuscript.

Reply:Thank you for your valuable comments. We have adjusted the paragraphs of the whole article. Because there are too many words involved, there is no emphasis on yellow.

  1. Table 4: The W/C is water-to-cement or water-to-cementitious ratio? Why 0.43 was selected for this study?

Reply:Thank you for your valuable comments. The W/C is water-to-cementitious ratio. By consulting the relevant literature and a large number of experimental studies, we determined that the W/C was 0.43. We have explained it in Section 2.2 of the manuscript and emphasized it in yellow.

8.Using nanomaterials reduces the workability of mortar along with using fibres. How did the authors consider this point? The mini-slump and mixtures' workability should be constant when comparing the mechanical characteristics. The reviewer recommends reporting mini-slump reports in Table 2 and explaining how much SP was used for each mixture.

Reply:Thank you for your valuable comments. We have supplemented the operation process and test results of the fluidity test of multi-walled carbon nanotubes/polypropylene fiber cement-based materials in Section 2.3.1 and Section 3.1.1, and emphasized them with yellow. The addition of MWCNTs and PP fibers will reduce the fluidity of cement-based composite mixture. The content of MWCNTs is 0.05%, 0.1%, 0.15% and 0.2%, it can be decreased by 6.7 %, 13.3 %, 18.7 % and 20 % respectively compared with the T0. The content of PP fiber is 0.1 %, 0.2 %, 0.3 % and 0.4 %, the fluidity decreases by 8.0 %, 14.2 %, 19.6 % and 24.4 % respectively compared with the T0, and when the content of PP fiber is 0.4 %, the fluidity does not meet the requirements of the specification. Compared with T0, the fluidity of T9-T12 decreases by 9.3 %, 18.2 %, 22.7 % and 26.7 %, respectively. The fluidity of the composite material is negatively correlated with the content of MWCNTs, and the fluidity gradually decreases with the increase of the content of MWCNTs. This is due to the large specific surface area of MWCNTs. After adding MWCNTs, the total specific surface area of cement-based materials increases relatively, and the free water adsorbed on the surface of MWCNTs increases significantly, which makes the cement paste become sticky, and with the increase of the content, the fluidity of the composite material decreases more obviously. The addition of PP fiber has an adverse effect on the fluidity of the composites. When PP fiber is added to the cement-based material, a three-dimensional network structure is formed inside the mixture, which increases the friction between the aggregates. At the same time, there is a certain viscous effect between the PP fiber and the slurry, resulting in a decrease in the fluidity of the cement mixture. With the increase of fiber content, it is difficult to disperse in cement-based composites, and the distribution spacing of PP fibers becomes smaller. Therefore, the fibers are easily agglomerated during the mixing process, which has a certain hindrance to the fluidity of cement paste. In addition, the amount of water reducer was obtained by referring to the relevant literature and debugging according to the actual mix ratio.

9.Table 2: percentages of MWCNTs are cement replacement (mass), but fibres should be volume fraction percentages. Please mention this within the table.

Reply:Thank you for your valuable comments. We have modified Table 2 and highlighted it in yellow.

  1. The authors must explain why the mixtures T9 to T12 were designed. Changing PP and MWCNTs percentages cannot be properly analysed without statistical analysis.

Reply:Thank you for your valuable comments. We apologize for the unnecessary trouble caused by our mistake. T9-T12 are fixed 0.1wt % multi-walled carbon nanotube content, change the content of polypropylene fiber. We have modified it in the manuscript and emphasized it in yellow.

  1. Please improve the quality of Figures 3-5.

Reply:Thank you for your valuable comments. We apologize for the problem of image distortion after uploading, and we have modified the images involved.

12.General comment: improving the compressive strength of cement-based composites containing nanomaterials was severely reported by researchers; accordingly, there is some concern regarding the novelty of the study. Please explain.

Reply:Thank you for your valuable comments. As proposed in the introduction of the manuscript, although polypropylene fiber can improve the flexural and tensile properties of cement-based materials, due to the lack of hydrophilic groups in its molecular chain, it has poor bonding with the cement matrix, and cracks are prone to occur at the fiber-matrix interface, thereby reducing the reinforcement effect of polypropylene fiber. The size effect and filling effect of carbon nanotubes can effectively promote the hydration process of cement, fill holes, and make the matrix more dense, thereby effectively improving the mechanical properties and durability of the cement matrix. Therefore, we hope that by incorporating MWCNTs into PP fiber composites, we can make up for the adverse effects of PP fibers on composites, enhance the interfacial bonding ability between fiber cement-based composites, improve the density of composites, and thus have better mechanical properties and durability.

13.Please add some photos of samples before and after performing mechanical tests.

Reply:Thank you for your valuable comments. We have added pictures of the specimen to the manuscript.

14.The reviewer recommends analysing the database of mechanical tests using statistical software for ANOVA to determine the synergistic effects of nanomaterial and fiber.

Reply:Thank you for your valuable comments. We apologize for the unnecessary trouble caused by our mistake. T9-T12 are fixed 0.1wt % multi-walled carbon nanotube content, change the content of polypropylene fiber. We have modified it in the manuscript and highlighted it in yellow.

15.Please enhance the quality of Figures 6-7 & 12.

Reply:Thank you for your valuable comments. We apologize for the problem of image distortion after uploading, and we have modified the images involved.

16.Section of 3.2.2. Freeze-thaw cycle test: Please add some photos of samples before and after F/T cycles.

Reply:Thank you for your valuable comments. We have added pictures of related experiments to the manuscript.

Thank you very much for your time and kind consideration.

Best regards,

Yours sincerely,

Yuanzhao Chen

Reviewer 2 Report

Comments and Suggestions for Authors

The manuscript "Study on the Properties of Multi-walled Carbon Nano-tubes(MWCNTs)/polypropylene Fiber(PP Fiber) Cement-based Materials" reports on the evaluation of mechanical properties and durability of the cement-based materials incorporating MWCNTs/ PP fibers. The authors have used a well-documented strategy to obtain such materials, the quantities being well chosen, although not justifying them, and a number of quite good properties depending on the quantities chosen. 

I identified a series of weaknesses in the content of this work, as follows:

Abstract: the last two sentences are made by assumptions, some specific methods to support them is needed.

Introduction: It is not clear the problem addressed and the solution given by these materials in the first paragraph.

The novelty of this study must be highlighted. What advantages were sought in the content of the designed composites. Similar conditions have already been investigated.

The motivation to choose PP and MWCNTs should be better emphasized and their mixing.

Section 2. Fig. 1 must include the same images after mixing with MWCNTs.

Table 3. Is not clear enough the utility of the presented materials, the discussion in section 2.2. is ambiguous. A schematic representation with the chemical transformation of MWCNTs to bind PP fibers must be shown.

Table 4. some typos must be corrected.

Figure 2. The steps must be explained in the legend of the figure.

In Section 3 all figures must be improved, their resolution is too low.

Some statements are made by assumption: "This is due to the incorporation of MWCNTs to improve the internal structure of PP fiber composites and reduce the porosity of PP fiber composites. At the same time, due to the nucleation of MWCNTs, the hydration of cement is promoted, the bonding performance between fiber and matrix is enhanced, and the toughening effect of PP fiber is further improved." - how authors proved this? some references must be added in some cases. This section lacks comparisons with other reports in the literature.

Fig. 5- delete the word "scheme". Rephrase the legend of the figure.

Section 3.4. How authors explain the large porosity of the composites? It seems that the addition of MWCNTs/ PP was not all beneficial? What about the behavior in the presence of humidity?

There is a correlation between the composition - structure-properties? This aspect must be better emphasized at least in the Conclusion section.

Based on these suggestions I consider that a major revision of the manuscript is needed before the acceptance.

Comments on the Quality of English Language

Some errors must be corrected in the content and References.

Author Response

Dear Reviewer:

Thank you for your valuable comments, we have carried on the careful discussion and modification, the specific reply is as follows:

1.Abstract: the last two sentences are made by assumptions, some specific methods to support them is needed.

Reply:Thank you for your valuable comments. We have modified the last two sentences of the Abstract and emphasized them in yellow.

2.Introduction: It is not clear the problem addressed and the solution given by these materials in the first paragraph.

Reply:Thank you for your valuable comments. We have supplemented the description of the problems and solutions solved by the material in the last paragraph of the Introduction.

3.The novelty of this study must be highlighted. What advantages were sought in the content of the designed composites. Similar conditions have already been investigated. The motivation to choose PP and MWCNTs should be better emphasized and their mixing.

Reply:Thank you for your valuable comments. As proposed in the introduction of the manuscript, although polypropylene fiber can improve the flexural and tensile properties of cement-based materials, due to the lack of hydrophilic groups in its molecular chain, it has poor bonding with the cement matrix, and cracks are prone to occur at the fiber-matrix interface, thereby reducing the reinforcement effect of polypropylene fiber. The size effect and filling effect of carbon nanotubes can effectively promote the hydration process of cement, fill holes, and make the matrix more dense, thereby effectively improving the mechanical properties and durability of the cement matrix. Therefore, we hope that by incorporating MWCNTs into PP fiber composites, we can make up for the adverse effects of PP fibers on composites, enhance the interfacial bonding ability between fiber cement-based composites, improve the density of composites, and thus have better mechanical properties and durability.

4.Section 2. Fig. 1 must include the same images after mixing with MWCNTs.

Reply:Thank you for your valuable comments. In this manuscript, polypropylene fiber was not blended with multi-walled carbon nanotube suspension in advance. Instead, multi-walled carbon nanotube suspension was prepared first, and then polypropylene fiber, multi-walled carbon nanotube suspension, cement and fly ash were added to the stirrer. The picture of the multi-walled carbon nanotube suspension is shown in Fig.2.

5.Table 3. Is not clear enough the utility of the presented materials, the discussion in section 2.2. is ambiguous. A schematic representation with the chemical transformation of MWCNTs to bind PP fibers must be shown.

Reply:Thank you for your valuable comments. We apologize for the trouble caused by our unclear statement. We have described it in detail in Section 2.1 and highlighted it in yellow. In this manuscript, nano-coating technology is not used to attach multi-walled carbon nanotubes to polypropylene fibers.

6.Table 4. some typos must be corrected.

Reply:Thank you for your valuable comments. We apologize for the trouble caused by our mistakes. We have modified Table 4 and highlighted it in yellow.

7.Figure 2. The steps must be explained in the legend of the figure.

Reply:Thank you for your valuable comments. We have added the relevant steps in Figure 2.

8.In Section 3 all figures must be improved, their resolution is too low.

Reply:Thank you for your valuable comments. We apologize for the distortion caused by the file upload. We have corrected all the pictures in the manuscript.

9.Some statements are made by assumption: "This is due to the incorporation of MWCNTs to improve the internal structure of PP fiber composites and reduce the porosity of PP fiber composites. At the same time, due to the nucleation of MWCNTs, the hydration of cement is promoted, the bonding performance between fiber and matrix is enhanced, and the toughening effect of PP fiber is further improved." - how authors proved this? some references must be added in some cases. This section lacks comparisons with other reports in the literature.

Reply:Thank you for your valuable comments. We have modified and emphasized in yellow the conclusions that appear throughout Section 3.

10.Fig. 5- delete the word "scheme". Rephrase the legend of the figure.

Reply:Thank you for your valuable comments. We have modified Figure 5.

11.Section 3.4. How authors explain the large porosity of the composites? It seems that the addition of MWCNTs/ PP was not all beneficial? What about the behavior in the presence of humidity?

Reply:Thank you for your valuable comments. After the composite specimens of T0 group, T2 group, T6 group and T10 group were prepared, the samples with a size of about 1 ~ 2g were selected after 28 days of curing age. The samples were immersed in a container filled with anhydrous ethanol to terminate the hydration of the samples, and then the soaked samples were placed in a drying oven at 105 °C for drying. Mercury intrusion test was only performed on the treated samples. And it can be seen from Fig.16 that the porosity curve of T2 group was lower than that of T0 group, and gradually tended to 0 with the increase of pore size. At the same time, according to the Table 7, the pore structure parameters such as average pore size and median pore size of T2 group were lower than those of T0 group, which indicated that multi-walled carbon nanotubes could effectively change the pore structure of cement-based materials. By comparing T0 and T6, it can be seen that the addition of PP fiber has an adverse effect on the overall pore structure of the composite. However, compared with the T6 group, the pore structure parameters of the T10 group were improved to some extent, indicating that the incorporation of MWCNTs had a certain refinement effect on the pore size of the PP fiber composite. This is due to the filling effect of MWCNTs and the promotion of cement hydration. The improvement of the pore structure improves the compactness of the composite. In addition, we have modified the relevant description of Section 3.4 and the conclusion part and emphasized it with yellow.

Thank you very much for your time and kind consideration.

Best regards,

Yours sincerely,

Yuanzhao Chen

Round 2

Reviewer 1 Report

Comments and Suggestions for Authors

The authors appropriately improved the manuscript.

Reviewer 2 Report

Comments and Suggestions for Authors

The authors responded to all the suggestions and the paper is acceptable in this form.